# Artificial and Natural Water Bodies Change in China, 2000–2020

Yong Wang [1,2,3], Shanlong Lu [1,2,4,*], Feng Zi [3,*], Hailong Tang [5], Mingyang Li [1,2,4], Xinru Li [1,2,4], Chun Fang [1,2,4] and Harrison Odion Ikhumhen [6]

1 International Research Center of Big Data for Sustainable Development Goals, Beijing 100094, China; 20010103004@mail.hnust.edu.cn (Y.W.); limingyang201@mails.ucas.ac.cn (M.L.); lixr2318@mails.jlu.edu.cn (X.L.); fangchun21@mails.ucas.ac.cn (C.F.)
2 Key Laboratory of Digital Earth Science, Aerospace Information Research Institute, Chinese Academy of Sciences, Beijing 100094, China
3 School of Earth Sciences and Spatial Information Engineering, Hunan University of Science and Technology, Xiangtan 411201, China
4 College of Resources and Environment, University of Chinese Academy of Sciences, Beijing 100049, China
5 School of Geospatial Engineering and Science, Sun Yat-sen University, Zhuhai 519082, China; tanghlong5@mail2.sysu.edu.cn
6 Key Laboratory of Ministry of Education for Coastal Wetland Ecosystems, College of the Environment and Ecology, Xiamen University, Xiamen 361102, China; harryspk@yahoo.com
* Correspondence: lusl@aircas.ac.cn (S.L.); zifeng@hnust.edu.cn (F.Z.)

**Abstract:** Artificial and natural water bodies, such as reservoirs, ponds, rivers and lakes, are important components of water-related ecosystems; they are also important indicators of the impact of human activities and climate change on surface water resources. However, due to the global and regional lack of artificial and natural water bodies data sets, understanding of the changes in water-related ecosystems under the dual impact of human activities and climate change is limited and scientific and effective protection and restoration actions are restricted. In this paper, artificial and natural water bodies data sets for China are developed for the years 2000, 2005, 2010, 2015 and 2020 based on satellite remote sensing surface water and artificial water body location sample data sets. The characteristics and causes of the temporal and spatial distributions of the artificial and natural water bodies are also analyzed. The results revealed that the area of artificial and natural water bodies in China shows an overall increasing trend, with obvious differences in spatial distribution during the last 20 years, and that the fluctuation range of artificial water bodies is smaller than that of natural water bodies. This research is critical for understanding the composition and long-term changes in China's surface water system and for supporting and formulating scientific and rational strategies for water-related ecosystem protection and restoration.

**Keywords:** artificial water body; natural water body; reservoir; surface water; water-related ecosystem

## 1. Introduction

Surface water bodies (including rivers, lakes, reservoirs and ponds, etc.) are the core elements in agriculture, aquaculture, industrial production and aquatic and terrestrial eco-systems [1], and the changes they undergo are important indicators reflecting the impact of climate change and human activities on surface water resources [2]. In recent decades, important components of surface water, such as artificial water bodies comprising reservoirs, canals, fish farming ponds, mines, quarries, etc., have been increasing in several regions of the world due to the construction of new reservoirs, climate change and flood irrigation [3]. Research shows that the number of new reservoirs is soaring along the world's largest river systems, such as the Yangtze, Euphrates, Tigris and La Plata [4]. As important components that distinguish artificial water bodies from natural water bodies, reservoirs play a vital role in the formation of artificial water systems through damming across rivers [5] or through closed water-proof embankments partially or completely formed

of concrete or clay [6]. In addition to the use of dams for hydropower generation, industrial purposes and peak flood attenuation, nearly 50% of dams worldwide are used to store, regulate and transfer water for agricultural irrigation [7–9]. With all of these benefits, the construction of dams is also known to have a negative impact on surrounding rivers, forests and swamp ecosystems, as well as global sea levels [10–13]. With the growing demand for renewable energy, many countries are developing small hydropower plants to meet the sustainable development of renewable energy because the renewable energy generated by small hydropower plants, especially run-of-river hydropower plants, is considered the most reliable source of clean energy in terms of low carbon emissions and it is generally believed that such hydropower plants have little impact on the environment [14]. There is no doubt that solar energy also has great potential in terms of clean energy [15]. However, due to economic and technical constraints, the expansion of small hydropower stations seems to be a better choice for most countries to balance energy and environmental concerns [16]. As a result, there are at present more than 80,000 small hydropower plants in operation worldwide, with a potential number of more than 180,000 [17]. Frustratingly, studies have shown that even small run-of-river hydropower plants can still have many adverse ecological impacts, such as changes in river flow and water temperature, loss of vertical connectivity, habitat destruction, loss of sensitive species, and simplification of fish and macroinvertebrate community composition [18–20]. These ecological impacts may be directly related to the lack of adequate environmental flow, regular monitoring and effective measures to ensure vertical connectivity [21]. It is therefore vitally important to research the distribution, changes and environmental–ecological effects of reservoirs and ponds representing artificial water bodies so as to objectively and accurately monitor and evaluate the changes in water-related ecosystems in different regions and carry out corresponding protection and restoration measures to achieve the global sustainable goals of SDG 6.6. However, the lack of a complete and authoritative artificial water body data set on a global scale, even one that includes large dams and reservoirs [22], has hindered the achievement of this goal, thereby influencing decision makers to formulate management measures for water eco-systems.

China has the world's largest population, the fastest growing economy, expanding irrigation and imperfect governance [23–25]. China possesses only 7% of the world's freshwater resources, and the per capita freshwater is below the world average [26]. The water and energy shortages experienced in China have prompted the Chinese government to build more dams and reservoirs [27]. According to the first China water resource census, since 2000, the construction of reservoirs nationwide has increased significantly [28]. As of 2011, China has built 97,246 reservoirs with a total storage capacity of more than $8.1 \times 10^{11}$ m$^3$, while the number of large dams has reached 23,842, accounting for about half of the world's registered large dams [29]. Moreover, this number is still increasing due to the demand for clean energy and potential hydropower resources [30]. The steady annual growth of reservoirs has led to complex physical and ecological changes in China's inland waters, most notably in the basin of the Yangtze River, the largest river in China, and there are increasingly more new reservoirs being constructed along the Mekong River (Lancang River). According to statistics, most of China's natural waters are regulated, with about 20% of the upstream areas of rivers (mainly in the north and northeast of China) having increased storage capacity in reservoirs [31]. However, this will inevitably lead to a variety of ecological problems, endangering ecological security and drinking water supply [32–34], so it is essential to know the detailed distribution of reservoirs and dams and reserved water in real time. Numerous studies have been conducted on surface water changes [35,36]; however, research on artificial water bodies, represented by reservoirs and ponds, is scarce and mostly limited to statistics of quantity and geographical location [37], ignoring spatial and temporal distribution characteristics [38]. For example, Wang et al. (2022) generated a data set of dams, reservoirs and lakes in China and comprehensively studied and recorded the spatial and temporal distribution characteristics of large dams,

reservoirs and lakes [39] but did not include other important artificial water bodies, such as medium and small-sized reservoir dams and artificial ponds.

In order to quantitatively characterize the quantity, distribution and spatiotemporal variation of artificial water bodies in China so as to provide data and decision-making support for the scientific development and utilization of artificial water bodies and the protection and restoration of natural water ecosystems, this study mainly focuses on two research aspects. The first concerns the development of Chinese artificial and natural water body data sets from 2000 to 2020 based on self-developed 30 m-resolution surface water data and artificial water body location data manually marked using the Google Earth Pro cloud platform by means of visual interpretation. The second involves analyzing the process and causes of spatiotemporal change in artificial and natural water bodies over the last 20 years.

## 2. Materials and Methods

### 2.1. Study Area

In this study, China was selected as the study area. China is located in eastern Asia (73°33′ E–135°2′ E, 3°52′ N–53°33′ N), and its topography is generally high in the west and low in the east, with highlands and mountains mostly in the west and plains in the east, roughly distributed in a three-stage gradient of which mountains, plateaus and hills account for about 67% of the land area and basins and plains for about 33% [40] (Figure 1). China's climate is complex and diverse, with monsoonal climates predominating in the northeast, continental climates in the northwest, highland and alpine climates on the Qinghai–Tibet Plateau and mainly tropical and sub-tropical climates in the south. The unique topography and complex climate lead to a gradual decrease in precipitation from southeast to northwest, while the temperature shows an opposite distribution, which determines the spatial distribution of water resources with a clear regional character [41]. In terms of overall distribution, water resources are more concentrated in the south than in the north. In order to study the differences in the distribution and variation of surface water bodies among different regions, nine sub-regions were divided according to previous studies [42]. These regions included the Southeastern River Basins (SERs), the Haihe River Basin (HR), the Huaihe River Basin (HuR), the Yellow River Basin (YR), the Northwest Inland River Basin (NWR), the Songliao River Basin (SLR), the Southwest River Basins (SWR), the Yangtze River Basin (YTR) and the Pearl River Basin (PR). The north–south variability in river runoff in China is obvious, with the five major basins in the north (SLR, HR, HuR, HR and NWR) accounting for less than 20% of the national land, while the four major basins in the south (YTR, PR, SWR and SER) account for more than 80% of the national land [43].

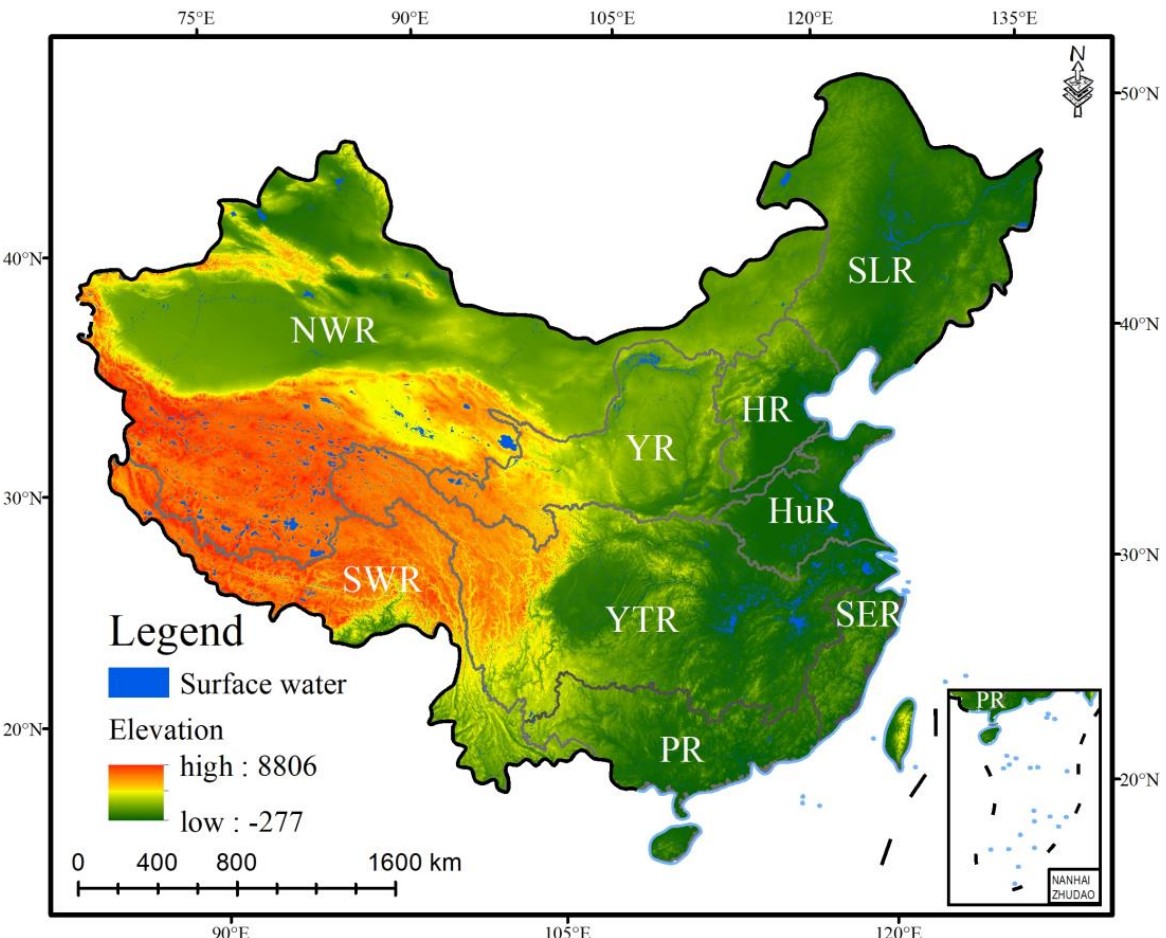

**Figure 1.** Spatial distribution and zoning of surface water in China. (SER: Southeast River Basin, HR: Hai River Basin, HuR: Huai River Basin, YR: Yellow River Basin, NWR: Northwest Inland River Basin, SWR: Southwest River Basins, SLR: Songliao River Basin, YTR: Yangtze River Basin, PR: Pearl River Basin.).

*2.2. Data Sources*

The research data used in this paper can be divided into three main classes: (1) surface water remote sensing monitoring data; (2) artificial water body distributing location data; and (3) longitude and latitude coordinate data of dams.

(1) Surface water remote sensing monitoring data. The surface water data are derived from the monthly surface water distribution data set (2000–2020) extracted from Landsat TM/ETM+/OLI imagery by our research team [44]. The European Commission's Joint Research Centre (JRC) global water surface distribution data (JRC Yearly Water Classification History, v1.3) from Google Earth Engine (GEE) was used for comparative validation.

(2) Sample data for artificial water body distribution. Based on GF-1/6, sentinel-2 and Landsat TM/ETM+/OLI images taken in 2019 and 2020, 45,585 artificial water body location points were manually annotated by visual interpretation.

(3) Dam latitude and longitude coordinate data. These mainly include China-IWRHR: dam location data provided by the China Institute of Water Resources and Hydropower Research; China-LDRL: the China Open Data Set on Large Dams, Reservoirs and Lakes developed and freely shared by Wang et al. [39]; GRanD v1.3: the Global Reservoir and Dam Database curated and hosted by Global Dam Watch [45]; and AQUASTAT: a global georeferenced database of dams collected by the Food and

Agriculture Organization of the United Nations (FAO) global information system on water resources and agricultural water management [46].

*2.3. Methodology*

Firstly, this study was based on a self-developed China Monthly Surface Water Data set (China-MSWD) using the monthly maximum synthesis method to obtain a China Annual Surface Water Data Set (China-ASWD) for each five-year interval from 2000 to 2020. Subsequently, the data set of artificial and natural water bodies in China for the years 2000, 2005, 2010, 2015 and 2020 were obtained through spatial overlay and segmentation by combining the data for 45,585 artificial water body points manually annotated by visual interpretation. Finally, the accuracy was verified using referenced dam data sources, and the characteristics and causes of spatiotemporal variation in artificial and natural water bodies in China from 2000 to 2020 were analyzed with reference to the monitoring method of SDG 6.6.1. A diagram of the specific research method is shown in Figure 2.

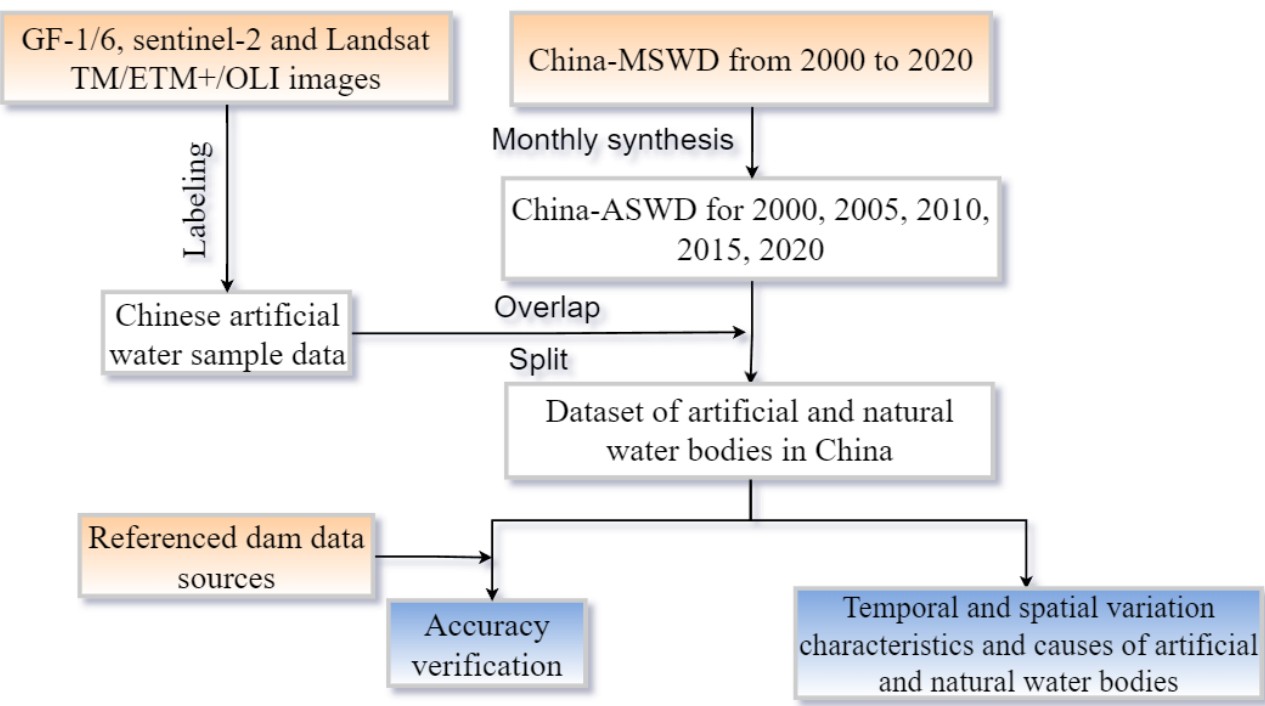

**Figure 2.** Diagram of the research method.

(1)　Synthesis of annual water surface data sets

Based on the China-MSWD with 30 m spatial resolution from 2000 to 2020, the China-ASWD for 2000, 2005, 2010, 2015 and 2020 was obtained through the synthesis of monthly maximum values. The calculation formulae are as follows:

$$ASW = MAX \{msw_1, msw_2, \ldots\ldots, msw_i\} \ (i = 1, 2, \ldots\ldots, 12) \tag{1}$$

where *ASW* is the annual surface water data in $km^2$ and $msw_i$ is the surface water data of the ith month in $km^2$.

(2)　Classification of artificial and natural water bodies

Using the 45,585 artificial water body data marked by visual observation as inputs, a spatial overlay analysis was carried out between the surface water data for each year and the selected artificial water body location samples. The results were used to determine the preliminary classification of artificial and natural water bodies. Afterwards, the natural

channels and artificial water bodies (including cascade power stations and reservoirs) were segmented manually to obtain the final artificial and natural water body data sets.

(3)  Construction of artificial and natural water body change indicators

Changes in the artificial and natural water bodies were characterized using the SDG6.6.1 indicators suggested by the United Nations. The calculation formula is as follows:

$$P = (\gamma - \beta)/\beta \tag{2}$$

where $P$ is the percentage change in the area of artificial or natural water bodies in different regions, $\beta$ is the average area of artificial or natural water bodies in different regions from 2000 to 2005 and $\gamma$ is the average area of artificial or natural water bodies from 2015 to 2020.

(4)  Spatial and temporal variation characteristics and causes of artificial and natural water bodies

By taking the provinces and river basins as units, the temporal and spatial variation characteristics of artificial and natural water bodies in China from 2000 to 2020 were analyzed. Considering that changes in artificial water bodies can be directly used to characterize the degree of human intervention in surface water resources, the geometric center of gravity analysis model in ArcGIS was used to reflect the spatial changes in an artificial water body's center of gravity over the past 20 years. The specific principles, concepts and calculation formulas are as shown in Equations (3)–(6).

First, the calculation formula of the geometric center of gravity is as follows:

$$\overline{x} = \frac{\sum_{i=1}^{n} x_i}{n}, \ \overline{y} = \frac{\sum_{i=1}^{n} y_i}{n} \tag{3}$$

where $n$ is the total number of spatial objects, $x_i$ and $y_i$ are the coordinate values of the ith spatial unit, $\overline{x}$ and $\overline{y}$ are the coordinate values of the geometric center. The formula for the area center of gravity model is as follows:

$$X = \frac{\sum_{i=1}^{n} s_i \cdot \overline{x}_i}{\sum_{i=1}^{n} s_i}, \ Y = \frac{\sum_{i=1}^{n} s_i \cdot \overline{y}_i}{\sum_{i=1}^{n} s_i} \tag{4}$$

where $X$ and $Y$ are the longitude and latitude values of the barycentric coordinates of the area attribute, respectively, $\overline{x}_i$ and $\overline{y}_i$ are the geometric center coordinates of the $i$th artificial water body area, and $n$ is the number of artificial water bodies. The formula for calculating the direction of the interannual spatial movement of the center of gravity is as follows:

$$\theta_{j-i} = \frac{n}{2}\pi + \arctan\left[\frac{y_j - y_i}{x_j - x_i}\right] \ (n = 0, 1, 2) \tag{5}$$

where $\theta_{j-i}$ represents the angle at which the center of gravity moves from the $i$th year to the $j$th year ($-180° \leq \theta_{j-i} \leq 180°$) and counterclockwise rotation is positive while clockwise rotation is negative. When the center of gravity shifts to the northeast direction, $0° < \theta < 90°$; when the center of gravity shifts to the northwest direction, $90° < \theta < 180°$; when the center of gravity shifts to the southwest direction, $-180° < \theta < -90°$; when the center of gravity shifts to the southeast, $-90° < \theta < 0°$; when the center of gravity shifts due east or due west, $\theta = 0°$ or $\theta = \pm180°$; and when the center of gravity shifts due north or due south, $\theta = \pm90°$. The formula for calculating the interannual spatial movement distance of the center of gravity is as follows:

$$D_{j-i} = c\sqrt{(X_j - X_i)^2 + (Y_j - Y_i)^2} \tag{6}$$

where $D_{j-i}$ represents the distance that the center of gravity moves from the $i$th year to the $j$th year and $c$ is the coefficient of converting the geographic coordinate system (°) to the plane projection coordinate system (km), $c = 111.111$.

## 3. Results

### 3.1. Accuracy Evaluation

#### 3.1.1. Accuracy Evaluation of Surface Water Data Sets

In order to assess the accuracy of the surface water data sets used in the study, the 2019 China ASWD and JRC annual water body data (including data for seasonal and permanent water bodies) were randomly selected at a spatial scale for comparison in four regions: the Qinghai–Tibet Plateau Lake Complex (A), the Lower Pearl River (B), Poyang Lake (C) and the Middle Songliao River (D). Among them, consistent, Under-extracted and Over-extracted in Table 1 represent the same, decreased and increased area of China-ASWD in 2019 compared to the JRC annual water data, respectively, and the matching rate represents the matching accuracy of the two data sets, which is equal to the ratio of the Consistent to JRC area. The result reveals that the water extraction accuracy of China-ASWD in the lake area is as high as 97% compared with the JRC data. Although the extraction accuracy of river channels, especially those near urban areas, was relatively poor, it was also observed to be above 90% (Figure 3, Table 1). Based on the time scale, the China-ASWD data also showed similar variation characteristics and good agreement with the JRC data, $R^2 = 0.6784$; furthermore, the extraction accuracy of China-ASWD was observed to be higher than that of the JRC data (Figure 4). Based on our other research, this was mainly because China-ASWD was more effective in the extraction of farmland and other plain areas with many small water bodies [44].

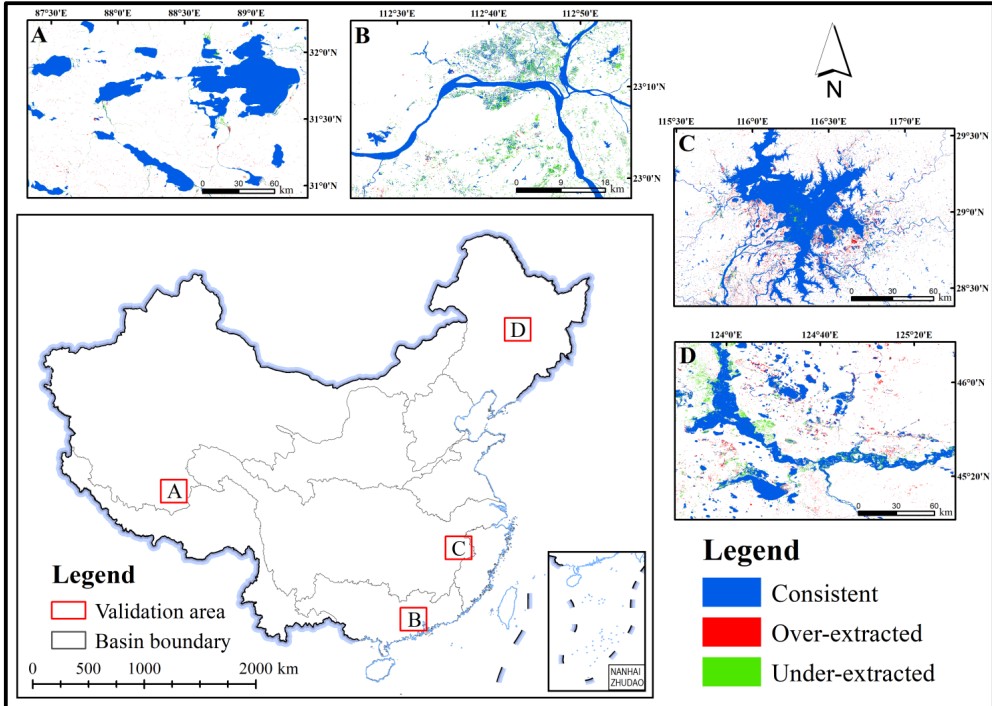

**Figure 3.** The consistent, over-extracted and under-extracted parts of the China-ASWD compared to the JRC data for regions (**A–D**) in 2019.

**Table 1.** Spatial comparison of China-ASWD compared to JRC water body data in 2019 for regions A, B, C and D.

| Region | JRC (km²) | Consistent (km²) | Under-Extracted (km²) | Over-Extracted (km²) | Matching Rate (%) |
|--------|-----------|------------------|------------------------|----------------------|-------------------|
| A | 4686.81 | 4548.55 | 138.26 | 100.76 | 97.05 |
| B | 252.91 | 230.82 | 22.08 | 9.00 | 91.27 |
| C | 4431.94 | 4314.65 | 117.29 | 415.96 | 97.35 |
| D | 3174.11 | 2964.07 | 210.04 | 316.86 | 93.38 |

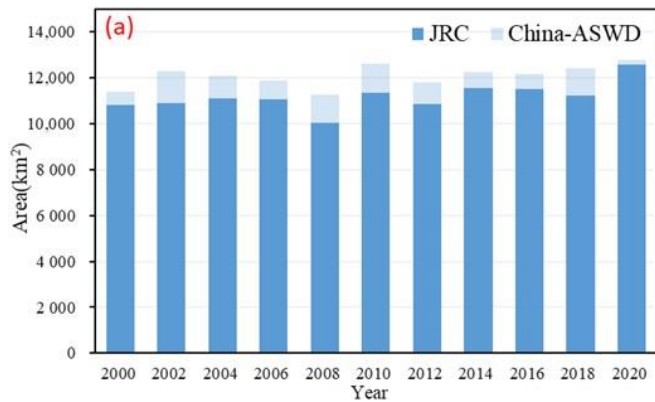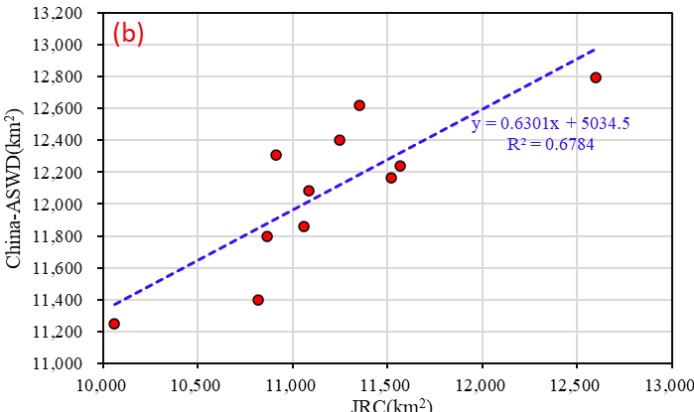

**Figure 4.** Time series comparison (**a**) and linear fitting result (**b**) of China-ASWD and JRC water body data for 2000–2020.

### 3.1.2. Accuracy Verification of Artificial Water Body Data Set

Since the existing data source only contains data for dam location points, when verifying the extraction accuracy of the artificial water body distribution data set, the dam buffer zone (centered on the dam location point and extending outwards 200 m) was generated according to the dam location data from different sources and the extraction accuracy of artificial water bodies was characterized by comparing the ratio of the buffer data and the number of water bodies intersecting China's artificial water bodies in 2020 to the number of all reservoir dam sample points. The results showed that the matching accuracies of the 2020 artificial water body data generated in this study with China-IWRHR, China-LDRL, GRanD v1.3 and AQUASTAT were 90.1%, 85.5%, 90.2% and 92.2%, respectively, with an average matching accuracy of 89.50% (Table 2).

**Table 2.** Comparison and verification of China's artificial water body data and other dam location data sources in 2020.

| Data Set | Total Dams | Dams in China | Matching Dams | Precision | Average Precision |
|----------|------------|---------------|---------------|-----------|-------------------|
| China-IWRHR | 4662 | 4662 | 4200 | 90.1% | |
| China-LDRL | 2140 | 2140 | 1830 | 85.5% | 89.5% |
| GRanD v1.3 | 7320 | 921 | 831 | 90.2% | |
| AQUASTAT | 14,500 | 722 | 666 | 92.2% | |

### 3.2. Spatial and Temporal Evolution of Artificial and Natural Water Bodies

Over the past 20 years, the area of natural water bodies in China has fluctuated greatly, showing an overall upward trend; contrary to that, the area of artificial water bodies has shown a continuous increase. During the 20-year study period, the total water body area has had a growth rate of 8.9% and increased by 18,245 km², natural water bodies representing the greatest proportion (68%), while artificial water bodies represent the smallest (31.5%) (Figure 5). In order to express the changes in the area of artificial and natural water bodies more clearly, we also adopted the Mann–Kendall (MK) trend test

based on $\alpha = 0.1$ to detect whether there had been an increasing trend in the area of artificial and natural water bodies over the past 20 years. Kendall's tau indicates the extent of the upward trend in the series data, which is positively correlated. The results showed that the area of artificial water bodies passed the MK test, while the area of natural water bodies did not. The rising trend in the area of artificial water bodies was significantly larger than that of natural water bodies (Kendall's tau: 0.8 > 0.2) (Table 3). In addition, based on the spatial distribution analysis conducted, it was observed that the natural water bodies were mainly distributed around the Qinghai–Tibet Plateau and the eastern and northeastern plains, while the artificial water bodies were observed to be mainly distributed in the middle and lower reaches of the Yangtze River and in coastal provinces (Figure 6).

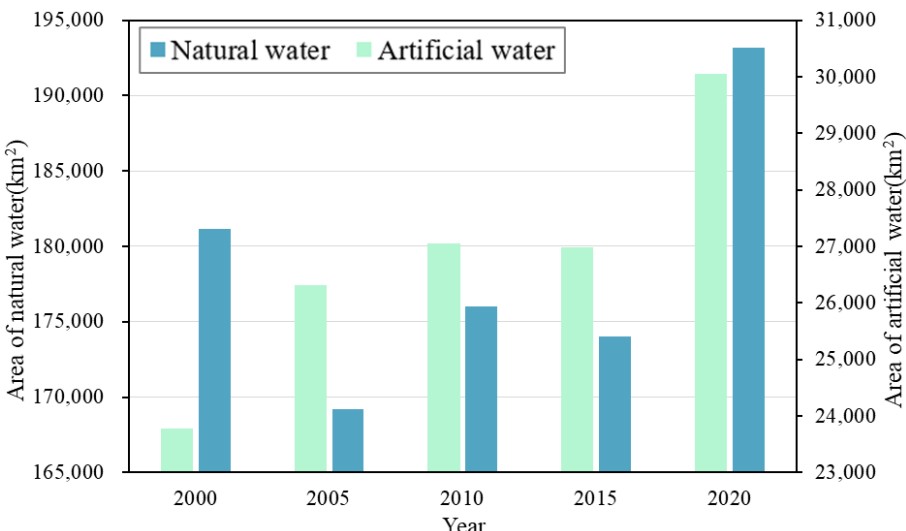

**Figure 5.** Changes in artificial and natural water bodies in China from 2000 to 2020.

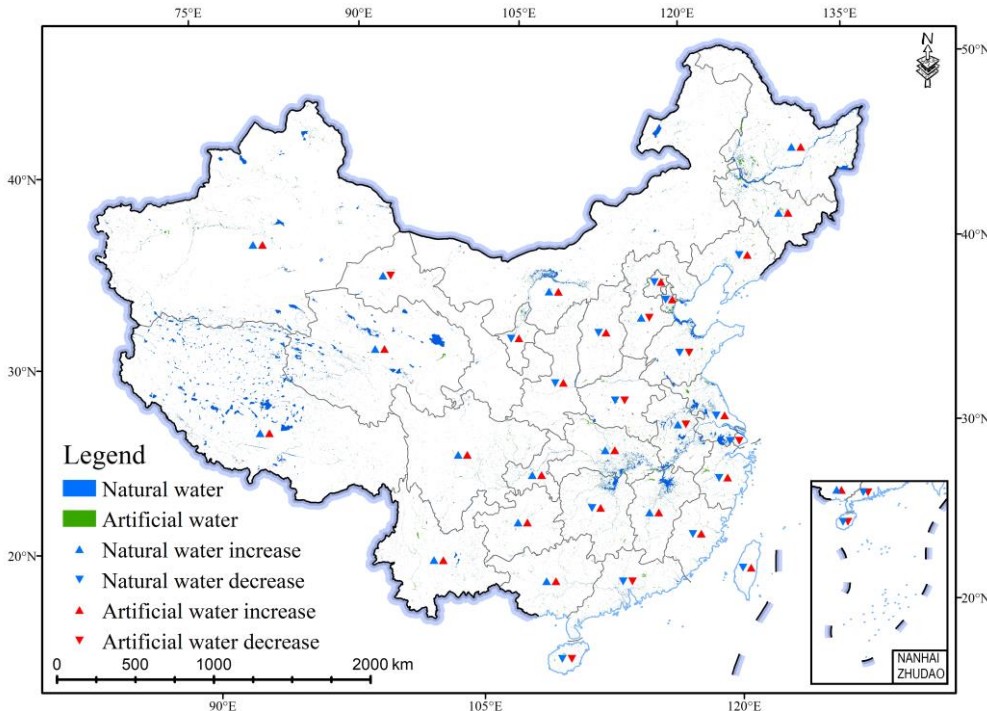

**Figure 6.** The spatial distribution of artificial and natural water bodies in China in 2020 and the change trend of artificial and natural water bodies in each provincial administrative region from 2015 to 2020 compared with 2000 to 2005.

**Table 3.** MK test results for the area of artificial and natural water bodies, 2000–2020.

| MK Test Parameters | Artificial Water | Natural Water |
| --- | --- | --- |
| Kendall's tau | 0.800 | 0.200 |
| *p*-value | 0.086 | 0.806 |
| Alpha($\alpha$) | 0.10 | 0.10 |
| $H_0$ | Reject | Accept |
| MK null hypothesis $H_0$: There is no trend in the series. | | |

With respect to provincial administrative units, the change trend analysis conducted between 2000–2005 and 2015–2020 revealed that there was an increment in the area of natural water bodies in 15 of China's 34 provincial administrative regions, including Guizhou, Anhui and Sichuan, etc. Nineteen regions, including Macau, Beijing, Tianjin, etc., observed a decrement in area. Furthermore, the artificial water bodies also experienced areal increase in 22 regions, including Yunnan, Guizhou and Hubei, etc., and areal decrease in 12 administrative regions, including Hebei, Shanghai, Anhui, etc. (Figure 6).

From the perspective of each river basin, the 2015–2020 data when compared with the data for the base period 2000–2005 revealed that, among the nine major river basins in China, the natural water body area of five river basins, including YR, YTR, PR, SLR and NWR, increased, accounting for 56% of the total number of river basins. Among them, the area of natural water bodies in YR and NWR increased significantly, with percentage changes of 11.3% and 11.2%, respectively. Additionally, the area of artificial water bodies increased in six river basins, including YTR, PR, SLR, NWR, SWR and SER, accounting for 67% of the total number of river basins. Other than the PR, there was a clear trend of growth in the area of artificial water bodies in other basins, especially in SWR, which exhibited the largest growth with a percentage change of 106.7%; this was followed by SLR which had a percentage change of 25.6%. The area of artificial water bodies decreased in three basins, including YR, HR and HuR, accounting for 33% of the total number of basins. Among them, the area of artificial water bodies in HR underwent the greatest decrement, with a percentage change of −16.7%; this was accompanied by the YR, which had a percentage change of −7.5%. HR was observed to be the basin with the smallest decrement in area, with a percentage change of −3.2% (Figure 7, Table 4).

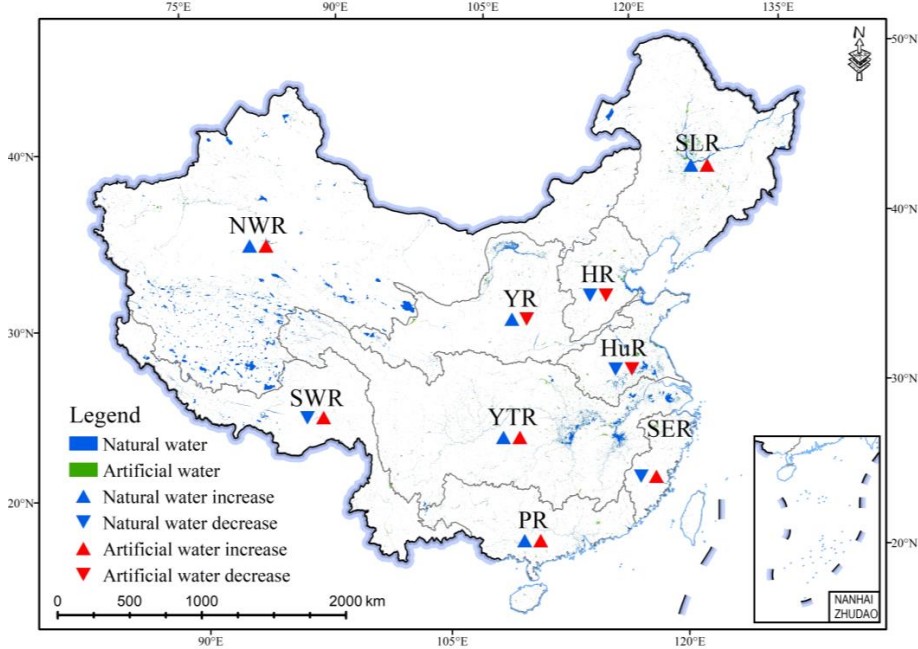

**Figure 7.** Spatial distribution of artificial and natural water bodies in China in 2020 and the change trend of artificial and natural water bodies in each basin from 2015 to 2020 compared with 2000 to 2005.

**Table 4.** Variation trends of artificial and natural water bodies in each basin from 2015 to 2020 compared with 2000 to 2005.

| Region | Type | $\beta$ (km$^2$) | $\gamma$ (km$^2$) | $P$ (%) |
|---|---|---|---|---|
| SER | Artificial | 1404.79 | 1586.99 | 13.0 |
| | Natural | 2459.98 | 2411.52 | −2.0 |
| HR | Artificial | 1757.57 | 1701.03 | −3.2 |
| | Natural | 5684.63 | 5220.06 | −8.2 |
| HuR | Artificial | 2459.65 | 2047.92 | −16.7 |
| | Natural | 11,383.76 | 10,980.19 | −3.5 |
| YR | Artificial | 2056.57 | 1901.43 | −7.5 |
| | Natural | 13,259.74 | 14,751.92 | 11.3 |
| NWR | Artificial | 2124.99 | 2409.33 | 13.4 |
| | Natural | 58,971.60 | 65,569.37 | 11.2 |
| SLR | Artificial | 5072.70 | 6370.03 | 25.6 |
| | Natural | 20,320.17 | 20,726.47 | 2.0 |
| SWR | Artificial | 416.11 | 859.98 | 106.7 |
| | Natural | 9683.71 | 9222.60 | −4.8 |
| YTR | Artificial | 7542.90 | 8023.30 | 6.4 |
| | Natural | 40,234.77 | 41,831.63 | 4.0 |
| PR | Artificial | 2758.85 | 2848.74 | 3.3 |
| | Natural | 5882.33 | 6023.87 | 2.4 |

As far as the fluctuation range is concerned, while the natural water area of other basins such as SWR showed limited fluctuation ranges, great fluctuations were observed for the HuR, YTR, PR, HR and SER basins. The smallest fluctuation specifically experienced in the natural water body area of SWR during the 20-year period under study is believed to be the result of the rare catastrophic drought in the five southwestern provinces of China in 2010. Similarly, severe drought and little precipitation in the northeast region during the springs of 2010 and 2015 were observed to be the factor responsible for the slight reduction in fluctuation range experienced in the natural water area of the SLR. Unlike the natural water bodies PR, YTR, and SWR, the areas of artificial water bodies in various river basins in the last 20 years showed general increasing trends (Figure 8).

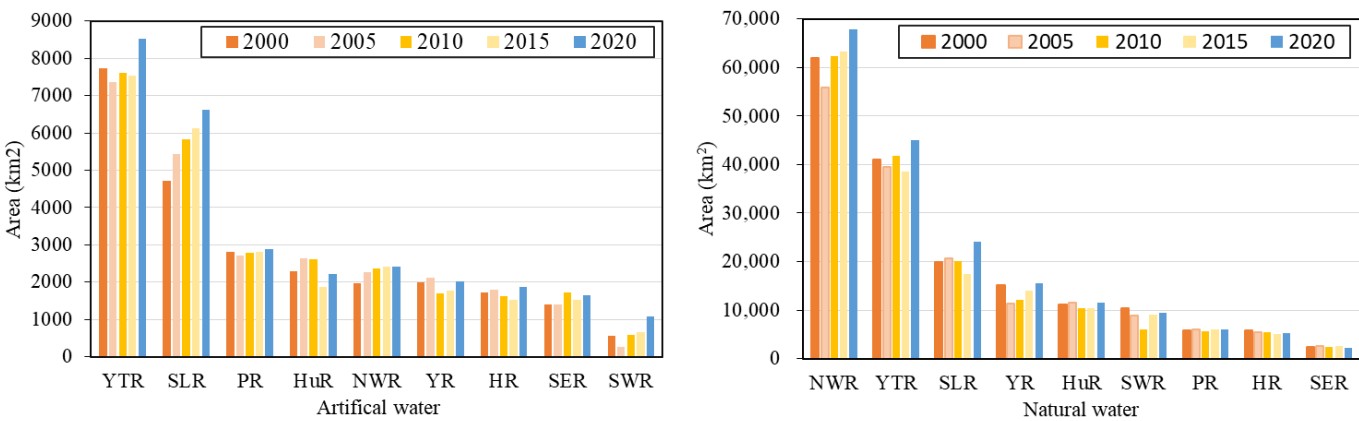

**Figure 8.** Changes in the areas of artificial and natural water bodies in each basin from 2000 to 2020.

*3.3. Changes in the Geometric Center of Gravity of Artificial Water Bodies*

Based on the area of artificial water bodies in each basin in China, the geometric center of gravity analysis model was used to obtain an overall movement direction and movement trajectory distribution map of the spatial distribution of artificial water bodies in China and each basin in the past 20 years (Figure S1, Table S1).

The results showed that, from 2000 to 2020, the distribution center of artificial water bodies in China was located between 113.074° E–113.275° E, 34.087° N–34.785° N, and that it was specifically distributed in Pingdingshan and Xuchang City of Zhengzhou, Henan

Province. The gravitational centers of artificial water bodies have been biased towards the east and south relative to the geometric center of China (103.842° E, 36.457° N, Lanzhou City, Gansu Province, China), which indicates that the distribution of artificial water bodies in China is in an unbalanced state, with distribution centered around the eastern and southern regions of China. According to the overall trend, China's artificial water distribution center of gravity in the last 20 years showed movements towards the northwestern region, with a total distance of 63.38 km. While the center of gravity moved about 0.34 km west in the east–west direction, with an average annual movement of 0.02 km, it moved about 63.37 km to the north in the north–south direction, with an average annual movement of 3.02 km. The movement velocity of the center of gravity of artificial water bodies in the north–south direction was much faster than in the east–west direction, indicating a significant growth in the northern region during the last 20 years.

The center of gravity of artificial water bodies in all basins, except NWR and the coastal basins of SWR, HR and HuR, moved in the direction of the basin's upper streams (Figure 9). This was believed to be due to the increase in the number of reservoirs in the middle and upper reaches of these basins over the last 20 years.

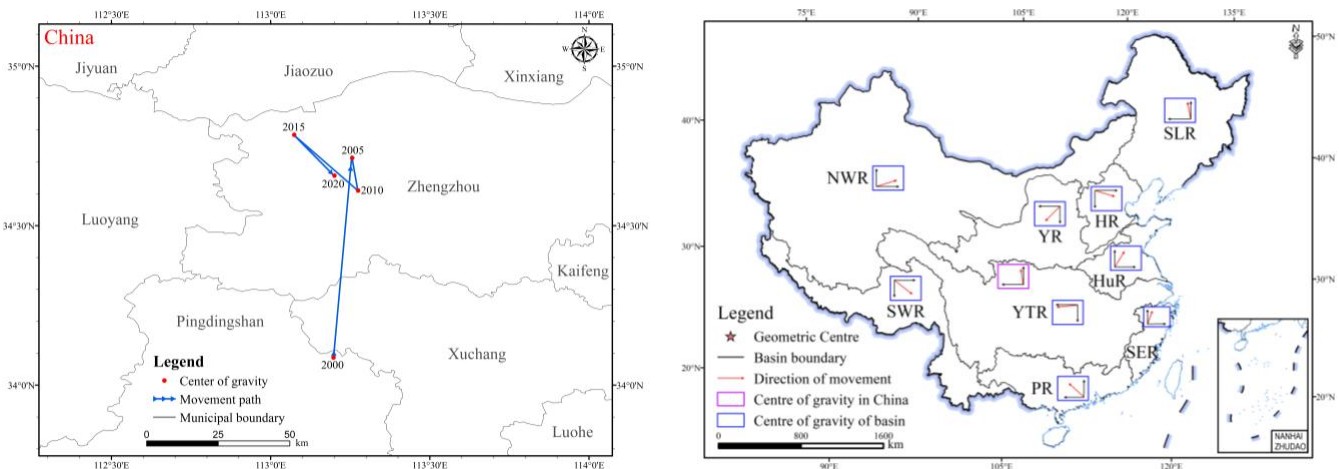

**Figure 9.** The movement trajectory of the center of gravity of artificial water bodies in China and various river basins from 2000 to 2020 and the direction of movement.

## 4. Discussion

### 4.1. Reasons for Changes in the Center of Gravity of the Distribution of Artificial Water Bodies in Typical Basins

To further investigate the reasons behind the shift in the center of gravity of artificial water body distributions in typical basins, a transitional change analysis was performed for artificial water bodies in SER, HR, HuR and NWR between 2000 and 2020 to evaluate the conditions of the artificial water bodies during this period.

The analysis results revealed that the completion of the Tankeng Reservoir (TK)/Qianxia Lake resulted in the gravity center of its artificial water body distribution shifting to the northeast of Lishui City, Zhejiang Province, by 12.39 km. The southeastward shift in the center of gravity experienced in the artificial water bodies (HR) is believed to be the result of an increment in the Bohai Rim Aquaculture (BRA) ponds, hence the shift in the gravitational center 6.28 km southeast. The artificial water bodies in HuR and HR showed a shift in centers of gravity in a similar direction (north east). This was believed to be a result of the expansion of plain reservoir construction in Shandong Province in the past 10 years [47]. Additionally, the Haizhou Bay Aquaculture (HBA) and other coastal aquaculture industries constructed for the bay area's economic development were observed to be the main causative factor in the gravitational shift towards the northeast, with a shifting distance of 31.11 km. In NWR, the predominance of artificial waterbodies with basically no large reservoirs or dams and the continuous advancement of the Heihe

Diversion Project (HDP) in Shaanxi Province is believed to be responsible for the significant shift in the gravitational center of the distribution of artificial water bodies 103.4 km to the northeast (Figure 10).

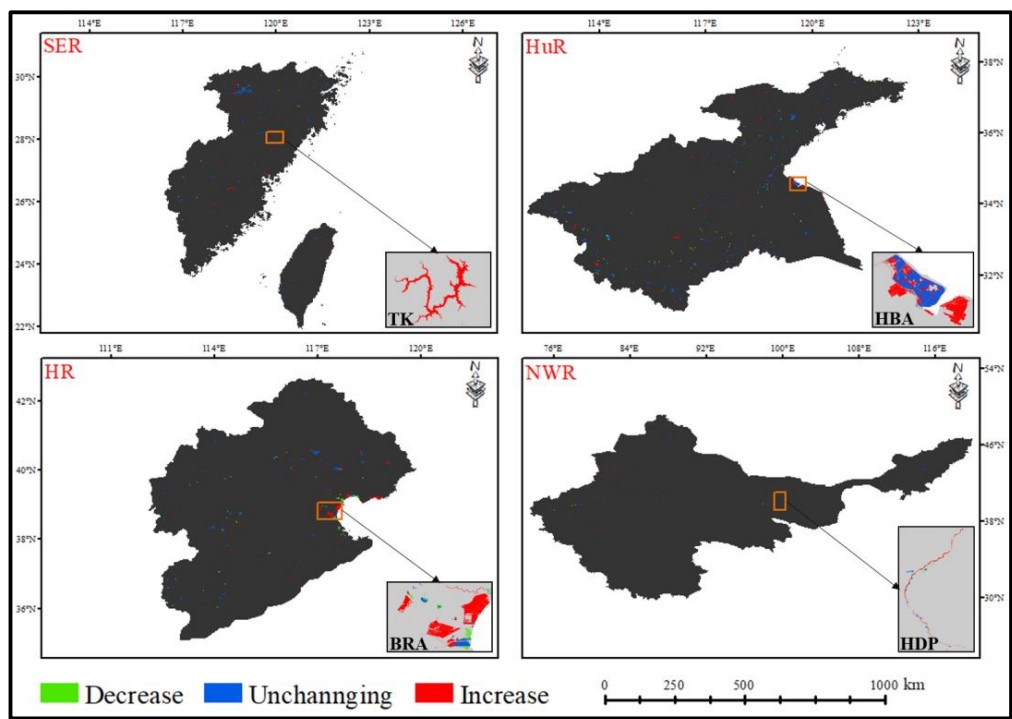

**Figure 10.** Changes in the spatial distribution of artificial water bodies in 2020 compared with 2000 in SER, HuR, HR and NWR.

*4.2. The Impact of Small and Medium-Sized Water Projects on Changes in the Distribution of Artificial Water Bodies*

Small and medium-sized reservoirs in China are characterized by their large number and wide distribution, yet previous studies have emphasized the impacts of large and very large dams while providing little basic information on smaller dams and the effects they cause [48,49]. Therefore, based on the results of this study, artificial water bodies were divided into three categories according to their size: small (<1 km$^2$), medium (1–100 km$^2$) and large (>100 km$^2$) [50].

The results show that in terms of quantity, small and medium-sized artificial water bodies in China accounted for more than 99.9% of the total of artificial water bodies, which was basically consistent with the situation of reservoir dams according to the First National Water Resources Census Bulletin [51]. In terms of area, small and medium-sized artificial water bodies in China account for about 80% of the total of artificial water bodies. In terms of temporal and spatial changes, it was easy to see that in the past 20 years there has been no obvious change in the proportion of small and medium-sized artificial water bodies in terms of area and quantity, with area increasing by just 1.1%, which indicates that China's small and medium-sized reservoirs are not only large in scale and long-term stability but also directly determine the distribution and changes in artificial water bodies in China. In terms of spatial distribution, small and medium-sized artificial water bodies in China were mostly found in YTR, SLR, PR and HuR, with YTR accounting for 39.46% of China's total small and medium-sized artificial water bodies and other river basins accounting for more than 10% (Figure 11). Despite the fact that these issues are receiving more attention, the lack of basic reliable data means that current knowledge is still based on studies of individual regions or a few large and medium-sized dams [52], which may not reveal the true scope and significance of large-scale impacts of various dams, especially small dams.

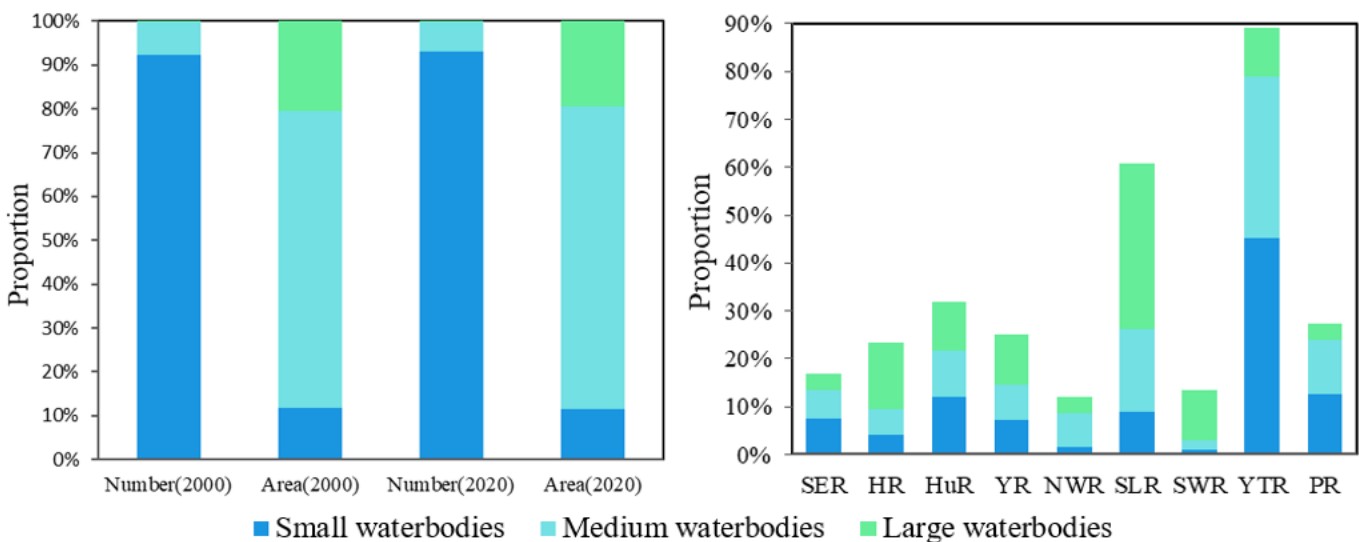

**Figure 11.** The proportion of artificial water body types in China in 2000 and 2020 and the proportion of artificial water body types in each basin in China in 2020.

The main differences between small and medium-sized dams and large dams are, firstly, their different hydrological effects. While large or very large dams (e.g., the Three Gorges Project) can store large amounts of water and have a considerable impact on the local hydrological cycle system, small dams also rely on their absolute numbers and densities to greatly influence the hydrological and ecological processes of river ecosystems and even the health of rivers [53–55]. Secondly, small and medium-sized dams have different uses to large dams. Fire protection, flood control and agricultural ponds are the main uses of small dams, while large dams are mainly used for residential water supply and hydroelectric power generation; the difference in uses suggests that cooperation with local people may be needed to mitigate and solve the environmental problems caused by small dams [56]. In summary, there is still a paucity of research on small and medium-sized artificial water bodies, both globally and in China, and more attention and research is needed for the management and construction of these large numbers of water bodies with significant local ecological impacts—for new clean drinking water projects [57], the demolition of functionally deficient water projects [58] and strengthening and de-risking of diseased and dangerous reservoirs [59] in order to achieve balanced social and ecological services in such artificial water ecosystems and maintain a healthy and sustainable state.

*4.3. Outlooks, Implications and Limitations*

Despite the importance and various roles of water-related ecosystems, they are still under great threat from human activities, mainly as a result of land conversion and the continuous transformation of natural water into artificial water bodies. The specific goal of SDG 6.6 is to provide the data necessary to understand the progress made and the problems that have risen and to provide basic knowledge about how to protect and restore these valuable ecosystems; however, surface water data alone are not enough to paint a full picture of the changes that have occurred. Therefore, the separation of artificial and natural water bodies is needed to help countries to better understand the value and benefits provided by water-related ecosystem services, in various sectors and societies, and to better assess the long-term impact of land use change and ultimately prioritize their restoration and conservation measures, especially for water source basins such as forests and critical watersheds. Currently, only the JRC produces a global reservoir dynamics data set with a spatial resolution of 30 m for representing surface area data of artificial water bodies, derived from the Global Surface Water Explorer (GSWE) data set, and separates artificial from natural water bodies by applying an expert system classifier to this data set [60]. However, it only documents 8869 reservoirs globally and explicitly states that

reservoirs built before 1984 with an area less than 30,000 m$^2$ and a width of less than 30 m are likely to be missing, so it falls far short of the global classification of artificial and natural water bodies. Against this background, this study has successfully generated a data set of artificial and natural water bodies in China using 45,585 manually annotated artificial water sample data and self-developed surface water data sets as inputs. The patterns of and reasons for change in the artificial and natural water area in China from 2000 to 2020 have been analyzed. Regional knowledge with respect to the SDG 6.6.1 indicators and the changes over time in this special surface water system have been obtained.

However, regarding the prospects of method application, there are two deficiencies in the surface water classification. Firstly, the important artificial water type of rice fields was not considered, as this type is difficult to capture accurately via single-phase remote sensing imaging. Secondly, the quality of remote sensing imagery used for visual interpretation of the artificial water bodies remains a problem when it comes to distinguishing artificial from natural water bodies, especially in hilly areas with complex topographies, such as southern and southwestern China. In addition, in terms of data analysis, the transformation from natural to artificial water bodies has disproportionate effects on local, regional and global carbon cycles and biodiversity levels [61–63], consideration of which is missing from the analysis in this paper due to methodological constraints.

## 5. Conclusions

This study made efforts to realize a secondary classification of surface water to compensate for the lack of global artificial and natural water body data sets. Furthermore, this study was able to successfully construct a data set and analyze the patterns and causes of spatiotemporal evolution in the distribution of artificial and natural water bodies in China for each five-year interval (from 2000 to 2020) with respect to provinces and river basins. The results indicated that, over the past 20 years, the area of artificial and natural water bodies in China has shown an overall increasing trend, with obvious differences in spatial distribution. Compared with natural water bodies, the fluctuation range of artificial water bodies was smaller. The center of gravity of artificial water body distribution in China was observed to shift northwards, while distribution in the east–west direction remained largely unchanged. The distribution centers of artificial water bodies in YR, YTR, PR, SLR and SWR were all observed to move in the direction upstream of the main rivers in the basin. Furthermore, several human activities, including the development of coastal fishing and farming in HR and HuR, the construction of large reservoirs (TK) in SER and large water-diversion projects (HDP) in NWR, were observed to be the main causative factors in relation to the shift in the geometric gravitational centers of these basins. Based on the categorization and spatial distribution change of artificial water bodies in China, it was noticed that small and medium-sized water bodies dominated the country, most of which were mainly located in the Yangtze River basin. Given the outcome of this study, we believe that the approach taken here could be a helpful tool in the improvement and optimization of research into patterns of surface water exploitation and use and that it could also assist the protection and restoration of surface waters and key water-related ecosystems. In the future, we will take this study as a foundation and combine machine learning algorithms to use higher-resolution remote sensing images to identify various artificial water bodies individually and seek to establish high temporal and spatial resolutions of artificial and natural water body data sets and actively promote them to other larger-scale regions in order to contribute to SDG 6.6.1 indicator monitoring and assessment capacity.

**Supplementary Materials:** The following supporting information can be downloaded at: https://www.mdpi.com/article/10.3390/w14111756/s1, Figure S1: The movement trajectory of the center of gravity of artificial water bodies in various river basins in China from 2000 to 2020; Table S1: The coordinates of the center of gravity, moving direction and moving distance of artificial water bodies in China and various river basins from 2000 to 2020; Table S2: List of acronyms used in this article.

**Author Contributions:** Conceptualization, S.L.; formal analysis, H.T.; data processing, H.T., Y.W., M.L., X.L. and C.F.; writing—original draft preparation, Y.W.; writing—review and editing, S.L. and H.O.I.; visualization, Y.W.; supervision, S.L. and F.Z.; funding acquisition, S.L. All authors have read and agreed to the published version of the manuscript.

**Funding:** The research was funded by the Strategic Priority Research Program of the Chinese Academy of Sciences (XDA19090120), the Second Tibetan Plateau Scientific Expedition and Research Program (STEP) (2019QZKK0202), the National Natural Science Foundation of China (42171283) and the National Key Research and Development Program of China (2021YFE0117800).

**Institutional Review Board Statement:** Not applicable.

**Informed Consent Statement:** Not applicable.

**Data Availability Statement:** Not applicable.

**Acknowledgments:** We thank Yunzhong Jiang from the China Institute of Water Resources and Hydropower Research for providing 4662 dam location data.

**Conflicts of Interest:** The authors declare no conflict of interest.

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
