# Peer review of "Artificial and Natural Water Bodies Change in China, 2000–2020"

_water, doi:10.3390/w14111756_

Round 1

Reviewer 1 Report

In the study, the authors analysed fluctuations in the number of artificial and natural water bodies in China from 2000 to 2020. First they presented the synthesis of annual water surface data sets, then the classification of artificial and natural water bodies and the construction of artificial and natural water body change indicators, and finally they analysed spatial and temporal variation characteristics and causes of artificial and natural water bodies.  Although the research is important for proper understanding the composition and long-term changes of China's surface water, I have a couple of remarks which may improve the manuscript. First of all,  rewrite abstract, improve the quality of figures, perform the trend analysis using statistical analysis and use appropriate statistical measures. Moreover, format references according to Water standards. Specific comments are given below.

Page 1, Abstract: rewrite the abstract presenting main research methods and results. Don’t repeat the period of your study (Lines 23 -24).

Page 2, Lines 83-91: please precise the aim and the scope of the article

Page 2, Lines 94-98:  please write references on mentioned topography

Page 3, Lines 99-112:  this information is very unprecise and vague. Please present brief information on climate and river basins and write references of course.

Page 3, Figure 1: Improve quality of the figure. Add geographical coordinates, the rectangle is vague.

Page 3, subsection 2.2: present detailed information on datasets used.  Explain all the abbreviations. Page 4, Line 140: In my view 2000-2020 should be

Page 4, Lines 142-144: what is the meaning of “monthly maximum values”. What about units: is it a number or surface area in km2? While writing about surface data present the unit. In section on research methods you use different grammar tenses.

Page 6, Line 212: In the determination coefficient statistically significant at α=0.05? What about the number of data?

Page 6, Figure 3: Improve quality of the figure. Enlarge geographical coordinates, the rectangle is vague. Specify areas A, B, C, D

Page 6, Table 1: What about the meaning and units for every column?

Page 7, Figure 4: Improve quality of the figure, add a) and b). I can’t find blue color for JRC.  What about statistical significance of the linear regression equation in the right figure?

Page 8, while you present the trend analysis, no quantitative information on results is presented. Carry out the trend analysis using statistical analysis and use appropriate statistical measures.

Page 8, Figure 6: Improve quality of the figure, in particular the rectangle is vague

Page 9, Figure 1: how did it increase significantly? Prove it giving  quantitative information.

Page 9, Figure 7: Improve quality of the figure, in particular the rectangle is vague

Page 11, Figure 9: Dreadful quality of the figure. Improve quality of the figure. Moreover in the caption “(taking China as an example)” is not necessary

Page 12, Figure 10: Improve quality of the figure.

Page 23: Format references according to Water standards.

Author Response

Thank you very much for reviewing our manuscript so carefull. All you comments and suggestions are so helpful for us to impove the manuscript. We have carefully considered them and revised one by one. Please find details in the following responses:

Reviewer 2 Report

This is an interesting study. Nevertheless, it needs some further improvements. In general, there are still some occasional grammar errors throughout the manuscript, especially the article "the," "a," and "an" is missing in many places; please make a spellchecking in addition to these minor issues. The reviewer has listed some specific comments that might help the authors further enhance the manuscript's quality.

  1. Specific Comments
  • A list of acronyms is needed

Introduction

  • The objectives should be more explicitly stated.
  • Please elaborate on the introduction section regarding how hydropower contributes to the renewable energy agenda also creating aartifical lakes and importance of e-flows. In this regard, the following literature may be helpful to <<Influence of hydrologically based environmental flow methods on flow alteration and energy production in a run-of-river hydropower plant>>, << Ecological impacts of run-of-river hydropower plants—Current status and future prospects on the brink of energy transition>>, <<Water spread mapping of multiple lakes using remote sensing and satellite data>> you may consider additional references as well. Also, I would suggest discussing different possibilities of usage of solar energy, such as one described in the following paper <<Energy Harvesting and Water Saving in Arid Regions via Solar PV Accommodation in Irrigation Canals>>
  • What is the novelty of this work?

  • Methods
  • The methodology limitation should be mentioned.
  • All variables should be explained.

  • Results
  • This section is well written.
  • I dont understand how it is possible the area of the natural lakes increase so much in such short time! Could you please explain it?

  • Discussion
  • Overall, the discussion part is week. The Discussion should summarize the manuscript's main finding(s) in the context of the broader scientific literature and address any limitations of the study or results that conflict with other published work.

Author Response

Thank you very much for reading our manuscript so carefully, and identifying and raising with us very constructive suggestions and comments. We have made revision one by one based on them, the following are our responses:

Round 2

Reviewer 1 Report

The authors corrected the text according to my expectations

Reviewer 2 Report

well done.